# Small Extracellular Vesicles as a New Class of Medicines

**DOI:** 10.3390/pharmaceutics15020325

**Published:** 2023-01-18

**Authors:** Inkyu Lee, Yoonjeong Choi, Dong-U Shin, Minjeong Kwon, Seohyun Kim, Hanul Jung, Gi-Hoon Nam, Minsu Kwon

**Affiliations:** 1KU-KIST Graduate School of Converging Science and Technology, Korea University, 145, Anam-ro, Seongbuk-gu, Seoul 02841, Republic of Korea; 2Department of Research and Development, SHIFTBIO Inc., Seoul 02751, Republic of Korea; 3Chemical & Biological Integrative Research Center, Biomedical Research Division, Korea Institute of Science and Technology (KIST), Seoul 02792, Republic of Korea; 4Department of Biochemistry and Molecular Biology, Korea University College of Medicine, Seoul 02841, Republic of Korea; 5Department of Otolaryngology, Asan Medical Center, University of Ulsan College of Medicine, Seoul 05505, Republic of Korea

**Keywords:** small extracellular vesicle, exosome, naïve small extracellular vesicle, engineered small extracellular vesicle, a new class of medicine

## Abstract

Extracellular vesicles (EVs) are nanovesicles that are naturally released from cells in a lipid bilayer-bound form. A subset population with a size of 200 nm, small EVs (sEVs), is enticing in many ways. Initially perceived as mere waste receptacles, sEVs have revealed other biological functions, such as cell-to-cell signal transduction and communication. Besides their notable biological functions, sEVs have profound advantages as future drug modalities: (i) excellent biocompatibility, (ii) high stability, and (iii) the potential to carry undruggable macromolecules as cargo. Indeed, many biopharmaceutical companies are utilizing sEVs, not only as diagnostic biomarkers but as therapeutic drugs. However, as all inchoate fields are challenging, there are limitations and hindrances in the clinical translation of sEV therapeutics. In this review, we summarize different types of sEV therapeutics, future improvements, and current strategies in large-scale production.

## 1. Introduction

sEVs are known to exert various functions, from aiding in the excretion of waste within cells to participating in cell-to-cell signal transduction [1]. Surprisingly, but importantly, sEVs have recently been found to play roles related to human diseases [2]. They disseminate diseases by transferring pathological cargo from abnormally altered cells to other cells. In this manner, sEVs derived from cancer cells can determine the aggressiveness of cancer or be associated with metastasis [2]. Furthermore, they have been found to be closely related to the pathophysiology of neurodegenerative and cardiovascular diseases [3,4]. Therefore, many studies are being conducted to exploit sEVs as diagnostic biomarkers and therapeutic targets by analyzing the proteins and nucleic acids in sEVs related to diseases [5].

Extracellular vesicles (EVs) are particles surrounded by lipid bilayers, naturally released from cells [2]. Although they are demarcated by lipid bilayers like cells, EVs cannot replicate and do not have a functional nucleus. Classifying EVs to certain biogenesis routes, such as the endosomal system (exosomes) or plasma membrane (ectosomes, microvesicles, microparticles), is challenging due to the lack of specific markers regarding each cell compartment [1]. Instead, EV subtypes are classified according to their physical properties (size or density), biochemical compositions (specific biomarkers), or recognizable conditions (cellular origin) [2]. For example, EVs with a size of up to 200 nm in diameter are classified as small EVs, and EVs larger than 200 nm are classified as medium/large EVs (m/lEVs). The Minimal Information for Studies of Extracellular Vesicles (MISEV or MISEV2018) encourages the use of small EVs (sEVs) instead of exosomes or microvesicles, with their size determined by appropriate methods [1]. Furthermore, EVs can be classified as low, middle, or high EVs according to their defined density range, or they can be distinguished through specific biomarkers, such as CD81^+^/CD9^+^ EVs. Lastly, nomenclatures based on cellular origins are often used, such as apoptotic bodies.

sEVs are currently receiving a great amount of attention as a promising therapeutic tool for diseases with high unmet medical needs due to their (i) excellent biocompatibility begetting low immunogenicity, (ii) high stability for the in vivo transport of substances, and (iii) the potential for loading a myriad of macromolecules as cargo [2]. Furthermore, sEVs perform the unique process of cellular signaling and uptake—they provide the optimal microenvironment for surface ligands to signal and display the intracellular delivery of therapeutic cargos to an extent [6]. Therefore, many scientists and drug developers are interested in the use of sEVs as a delivery tool for proteins and genes. In addition, sEVs show an advantage in cell-free therapies, as they can overcome the current safety concerns and challenges regarding the injection of viable cells and scalable manufacturing [2,6].

Although sEVs have shown versatility and high potential as a disease treatment at the pre-clinical level, limitations of the practical application in clinical practice remain [2]. Therefore, in this review, we introduce the methods and studies of sEVs being used as therapeutic agents for human diseases and discuss problems and notable points to be overcome in future clinical applications regarding sEV-based drugs.

## 2. Types of sEV Therapeutics

The current sEV therapeutics include utilizing naïve/engineered sEVs and suppressing the secretion/uptake of sEVs, as shown in Figure 1. Many researchers and biotech companies have been aiming to develop a new class of medicines harnessing sEVs harboring macromolecules [2,6]. They have used sEVs for cell-free therapy and as efficient delivery tools for therapeutic cargo, targeting high unmet medical needs. Furthermore, since various diseases have been found to progress through the communication of sEVs produced from transformed cells, some studies have suggested that the inhibition of the production and uptake of pathological sEVs could be a promising therapeutic strategy [2]. In this section, we address several sEV therapeutic strategies.

### 2.1. Inhibition of the Release and Uptake of sEVs

Emerging evidence points to sEVs having roles in human diseases. sEVs disseminate diseases by transferring pathological cargo from diseased donor cells to normal cells. For instance, cancer cell-derived sEVs are associated with tumor progression and metastasis [7,8,9] Therefore, it is important to understand the biogenesis of sEVs and strategies to inhibit the release and uptake of sEVs.

Intracellular sEV production is predominantly based on two pathways: endosomal sorting complexes required for transport machinery (ESCRT)-dependent and -independent pathways. In the former case, multi-vesicular bodies (MVBs) are formed by ESCRT, and intraluminal vesicles (ILVs) contained therein are released in the form of EVs outside the cell. In the ESCRT-independent pathway, MVBs-ILVs are formed by neutral sphingomyelinase 2 (nSmase2) through sphingomyelinase hydrolysis and ceramide formation [10]. Accordingly, drugs that inhibit sEV release mainly target ESCRT and nSmase2. ESCRT is recognized to be closely related to the rat sarcoma virus (Ras) family protein. Drugs such as manumycin A and tipifarnib are often used as therapeutic agents to reduce sEV secretion by inhibiting ESCRT due to their specific inhibition of farnesyltransferase, one of the major enzymes in the lipid metabolism pathway of Ras. Datta et al. reported that inhibiting Ras/Raf/MEK/ERK1/2 signaling using Manumacin A can suppress the biogenesis of sEVs in castration-resistant prostate cancer cells, and Greenberg et al. highlighted the reduction in sEV secretion in sunitinib-sensitive renal cell carcinoma when treated with Tipifarnib [11,12]. The nSMase2 is a ubiquitous enzyme that generates a bioactive lipid ceramide through the hydrolysis of the membrane lipid sphingomyelin. A potent and specific nSMase2 inhibitor, such as GW4869, can prevent the formation of ILVs and consequently diminish sEV production [13]. Similarly, Imipramine also affects the lipid metabolism of the secreting cells, thereby reducing micropinocytosis activity and decreasing the secretion of sEVs. Hekmatirad et al. reported that the prevention of sEV release by GW4869, the nSMase2 inhibitor, can enhance doxorubicin sensitivity U937 cells by the inhibition of expelling doxorubicin via sEVs [14]. Other drugs known to inhibit sEV release are summarized in Table 1.

Another strategy to inhibit the propagation of sEVs is to inhibit uptake in recipient cells. The primary mechanism of sEV uptake is associated with the endocytosis pathway, which is divided into clathrin-dependent and -independent mechanisms [15]. Furthermore, other sEV uptake processes are mediated through membrane fusion, phagocytosis, and micropinocytosis [15]. Dynasore, a dynamin inhibitor, was found to inhibit sEV uptake by interfering with the transferrin internalization through the clathrin-dependent pathway, and it has been reported to inhibit the angiogenesis associated with the propagation of sEVs derived from malignant melanoma cells [16]. In addition, Nanbo et al. reported that dynasore could hamper virus spread by inhibiting the uptake of exosomes derived from Epstein–Barr virus-infected B cells into uninfected B cells [17]. Heparin was reported to hamper the sEV delivery between doner and recipient cells via competitive binding to the receptors of recipient cells [18]. Therefore, the uptake of sEVs that involve heparin sulfate proteoglycan (HSPG) coreceptors can be prevented by pretreatment with heparin on the recipient cells [19,20]. Unfortunately, a therapeutic strategy to inhibit sEV uptake is not readily applicable due to the convoluted mechanisms and the unfeasibility of the selective inhibition and visual verification of sEV uptake in specific cells [21]. For instance, a method for selectively inhibiting pathological sEVs by distinguishing them from healthy sEVs essential to maintaining normal physiological functions has not been demonstrated to date.

**Table 1 pharmaceutics-15-00325-t001:** Therapeutic approaches via inhibition of the release and uptake of sEVs.

Strategy	Reagent	Disease	Mode of Action	Treatment	Result	Reference
Exosome biogenesis and release inhibition	Manumycin A	C42B prostate cancer	ERK-dependent inhibition of hnRNP H1	250 nM treatment for 48 h	Decreased Ras activation by GTPγS	[11]
	GW4869	U937 acute myeloblastic leukemia	Enhance doxorubicin sensitivity on resistant cells by inhibition doxorubicin expulsionvia sEVs	20 µM GW4869 + 0.5–2 uM PEGylated liposomal doxorubicin for 24 h	Enhanced cytotoxicity	[14]
	Tipifarnib	Metastatic renal cell carcinoma	Disrupt ESCRT-dependent and ESCRT-independent functional proteins (Alix, nSMase, and Rab27a)	0.25–1 µM treatment for 48 h at 37 °C	Reduced sEV load in sunitinib-sensitive renal cell carcinomas	[12]
	5-(*N*-ethyl-*N*-isopropyl) amiloride	A431 human epidermoid carcinoma	Disrupt Rac1 activation and assembly of actin	50–100 µM pretreatment for 30 min	Inhibited clathrin-independent endocytosis and macropinocytosis of sEVs	[22]
	Imipramine	PC3 prostate cancer	Reduce macropinocytosis	25 µM treatment	Reduced total EV release by 77% in PC3	[23]
		4T1 mammary carcinoma		5 μM treatment for 1 h	Inhibited membrane ruffle formation	[24]
	Calpeptin	Worms	Inhibit calpain	80 μM treatment	Prevented the secretion of miRNAs from adult worms	[25]
Exosome endocytosis inhibition	Dynasore	HUVEC	Interfere with the internalization of transferrin through the clathrin-dependent pathway	Pretreatment on HUVECs with 20 µM dynasore for 30 min at 37 °C	Prevented pancreatic cancer cell-derived sEVs	[16]
		B cells		Pretreatment on uninfected B cells with 150 μM dynasor for 30 min at 37 °C	Prevented virus spread by inhibiting the uptake of Epstein–Barr virus-infected B cell-derived exosomes	[17]
		Human bone marrow stromal cells		Pretreatment on bone marrow stromal cells with 50 μM dynasore for 30 min	Suppressed the effects of multiple myeloma cell-derived exosome uptake	[26]
	Heparin	U-87 MG glioblastoma	Competitively inhibit cancer cell surface receptors depending on heparin sulfate proteoglycan coreceptors for the uptake of sEVs	Pretreatment on U-87 MG with 10 μg/mL heparin for 1 h	Reduced sEV uptake in U87 cells by 55%	[19]
		SW780 human bladder cancer cell line		Pretreatment on SW780 with 10 μg/mL heparin at 4 °C for 4 h	Inhibited receptors on recipient cells	[20]

### 2.2. Naïve sEV Therapeutics

Naïve sEVs, or native sEVs, reflect diverse characteristics, such as membrane proteins or contents, of their parental cells. By leveraging this property, various studies reported the potential of naïve sEVs’ therapeutic efficacy. Depending on the origin of cells, sEVs can be utilized in appropriate diseases. In this section, we explore various therapeutic effects of naïve sEVs derived from stem cells, immune cells, and other cells, such as red blood cells or platelets.

#### 2.2.1. Stem Cell-Derived Naïve sEVs

Stem cells are undifferentiated but can be multilineage differentiated cells with self-renewal capability. Accordingly, stem cells have been frequently and widely used in clinics, especially in regenerative medicine, regarding their pleiotropic differentiating potential and immunomodulatory properties [27]. Generally, stem cells are divided into embryonic stem cells, adult stem cells that originate from diverse mesenchymal/stromal tissues and bone marrow, and induced pluripotent stem cells. A large body of work has demonstrated that mesenchymal stem cells (MSCs) can modulate inflammation and immune responses [27]. MSCs can be harvested from diverse tissues, including bone marrow, the umbilical cord, adipose tissue, and brain tissue. Notably, MSCs activated by inflammatory cytokines have tropism to ischemia, injury, or inflammation sites [28]. Furthermore, MSCs can suppress pro-inflammatory processes by releasing an array of factors [29]. These factors consist of interleukin-10 and growth factors (GFs), such as trans-forming GF-β (TGF-β), hepatocyte GF (HGF), stromal-cell-derived factor-1 (SDF-1), epidermal GF (EGF), keratinocyte GF (KGF)-1, fibroblast GF (FGF), vascular endothelial GF (VEGF), plate-derived GF (PDGF), and insulin GF (IGF)-1.

Nonetheless, stem cell therapies show several notable qualities regarding large-scale production, quality control, and off-the-shelf medicines [30]. Moreover, MSC-based therapeutics have been reported to elicit tumorigenicity as a severe side effect [30]. Interestingly, recent studies discovered that stem cells’ abilities arise from the secretion of paracrine factors, and sEVs comprise the primary mediators [31,32]. Therefore, MSC-derived sEVs offer the possibility of an alternative approach to stem cell therapeutics for treating various diseases, including cardiovascular, neurodegenerative, and immunologic diseases, as shown in Table 2. Zhao et al. reported that micro-RNA-182-containing MSC-derived sEVs could attenuate myocardial ischemia–reperfusion injury via changing M1-like polarized macrophages into M2 phenotypes [33]. Alzheimer’s disease (AD) is among the most problematic and frequently investigated human diseases in terms of neurogenerative disorders, and therefore, a number of attempts have been made to mitigate the progression of Altzheimer’s disease using MSC-derived sEVs. A recent study demonstrated that intranasally inhaled human MSC-derived sEVs could slow down AD-related pathogenesis [34]. Furthermore, MSC-derived sEVs could mitigate autoimmune-related demyelinating processes and influence neuroprotective mechanisms via systemically modulating regulatory T cells and peripheral blood mononuclear cells [35]. The normalization of kidney function can be achieved by the administration of stem cell-derived sEVs on chronic kidney disease or diabetic nephropathy [36,37]. These findings suggest the potential of MSC-derived sEVs as an alternative approach to stem cell therapies in diverse diseases.

#### 2.2.2. Immune Cell-Derived Naïve sEVs

According to previous reports, tumor cell-derived EVs (TEVs) have shown conflicting characteristics in terms of both promoting the aggressiveness of tumors and initiating anticancer immunity cycles. Zitvogel et al. reported the promising vaccine effects of TEVs as sources of tumor antigens for the first time [52]. This study found that TEVs induced better anti-tumor immune responses than tumor cell lysates. Moreover, TEVs can directly activate antigen-presenting cells (APCs) to release pro-inflammatory cytokines [53]. Notably, TEVs can elicit prophylactic cancer vaccine effects but not therapeutic ones [54]. Contrary to the anti-tumor immunity of TEVs, TEVs have been found to provoke heterogenous pro-tumorigenic effects dependent on the tumor type and stage, such as initiating cell transformation, modulating the tumor and metastatic microenvironment, and fostering tumor progression [55,56,57,58]. These results collectively indicate the ambivalence of TEVs in cancer vaccines, which represents a hurdle to their use as transformative medicines. Therefore, researchers have developed alternative methods, such as immune cell-derived sEVs, for the anti-tumor therapeutic approach.

sEVs derived from tumor antigen-exposed dendritic cells (DEVs, sometimes referred to as ‘dexosomes’) have been considered [59] to overcome the limitations of TEVs for cancer vaccines. This approach takes advantage of the molecular properties of DEVs, which contain the adhesion/docking molecules and immunostimulatory factors presented on DCs. Specifically, peptide–MHC complexes can be formed spontaneously and loaded at the external surface of DEVs, thereby provoking additional immunostimulatory effects [60]. Interestingly, DEVs produced more effective anticancer immunity than microvesicles from DCs [61]. Since DEVs show the efficient induction of tumor-specific immunity, their potential to be utilized as an anticancer vaccine has been investigated extensively, concurrent with multiple ongoing clinical trials, as recently reviewed [62]. The completed clinical trials on DEV-based therapy have reported mild to moderate side effects, with relatively milder responses such as low levels of T cell responses, with NK cell stimulation, showing tolerability among cancer patients [62]. According to the clinical trial on metastatic melanoma patients treated with DEVs containing MHC class II peptides as a vaccination, the majority of patients showed a minimal response, and only one patient showed the recruitment and activation of T cell response in the tumor area. Similarly, a modest response was reported on patients with non-small-cell lung cancer, with two patients showing enhanced activity in NK cells. Overall, the results from completed clinical trials imply that DEVs bear the potential to be utilized as cancer vaccines. Studies need to be further implemented or engineered to maximize and guarantee immune response enhancement.

Other than DCs, some studies have evaluated the anti-tumor effects of sEVs from immune cells. For instance, NK cell line-derived sEVs (NK-EVs) have been reported to eradicate specific cancer cells through cytotoxic molecules such as tumor necrosis factor-α, perforin, granzyme, and the Fas ligand [63,64,65]. Although the mechanism of M1-type macrophage-derived sEVs (M1-EVs) in increasing anticancer immunity is still unclear, several reports demonstrated that M1-EVs induced anti-tumor immune responses through activating APCs, including macrophages and dendritic cells [66,67]. Furthermore, sEVs derived from effector chimeric antigen receptor (CAR) T cells (CAR-EVs) have been shown to mainly carry CAR and sufficient cytotoxic molecules, without expressing programmed cell death protein-1 (PD-1) on their surfaces [68]. Notably, CAR-EVs substantially inhibited tumor growth without cytokine release syndrome, which is the primary side effect of CAR-T cell therapy, and the tumor-inhibiting efficacy did not decrease with the treatment of PD-1:PD-L1 blockade [69]. Collectively, these studies demonstrate the wide versatility of immune cell-derived sEVs as a promising therapeutic strategy for cancer treatment (Table 3).

#### 2.2.3. Other Cell-Derived Naïve sEVs

Platelets, or thrombocytes, are anuclear cells produced in the bone marrow. They were once regarded as mere fragments of megakaryocytes. However, accumulated research has pointed out the important biological roles of platelets, including angiogenesis, hemostasis, and thrombosis [74]. Platelet-derived EVs (pEVs or PLT-EVs) have been found to take the lead in these biological roles through intercellular communication. Most EVs isolated from human serum are pEVs, and they participate in both regenerative responses, such as wound healing or tissue regeneration, and pathological processes, including inflammation and tumor progression [75]. During the normal wound-healing process, it is unequivocal that keratinocytes and fibroblasts orchestrate most of the repair process [76]. Previous reports have demonstrated, in vitro, that growth factor cargos of pEVs influence these repair cells to migrate and accumulate in wound sites. These growth factors include platelet-derived growth factor (PDGF), basic fibroblast growth factors (FGF2), transforming growth factor-β (TGF-β), and vascular endothelial growth factor (VEGF) [76]. Guo et al. observed wound-healing effects of pEVs, enhancing cell proliferation and migration through an angiogenesis-promoting effect by stimulating HMEC-1 growth factor in diabetic rat models [76]. Furthermore, previous studies have shown another interesting property of pEVs having neuroregenerative effects [77]. Protein cargos such as PDGF and VEGF in pEVs influenced the neurogenesis of neural stem cells. Hayon et al. used permanent middle cerebral artery occlusion (MCAO) rat models to show a dosage-dependent increase in neuroregenerative effects of pEVs [77]. On the other hand, many researchers have studied pEVs that aggravate cancer progression or inflammation. Ironically, the critical crosstalk between MAPK and YAP during the wound-repair process arranged through pEVs is hijacked in tumor cells. Labelle et al. showed exacerbated metastasis through platelet-derived TGFB that activates the NF-kB pathway [78].

Red blood cell (RBC)-derived EVs (RBCEVs) have gained attention due to their safety and biocompatibility in clinical applications. For instance, RBCEVs have a lower risk of horizontal gene transfer, because they lack nuclear DNA and mitochondria. RBCEVs participate in important biological processes, such as nitric oxide homeostasis, redox balance, immunomodulation, and coagulation [79]. Hitherto, there is no gold standard on which cell type-derived EVs should be used for drug delivery systems due to their unique property of reflecting proteins of their donor cells. For instance, EVs derived from tumor cells pose tumorigenicity, and EVs from nucleated cells have the risk of horizontal gene transfer. However, cumulative research points to the potential and strong efficacy of RBCEVs as a drug delivery system. Zhang et al. showed the feasibility of RBCEVs as a drug delivery system in acute liver failure (ALF) mouse models by loading antisense oligonucleotides of miR-155 (miR155-ASO) [80]. Although the downregulation of miR-155 through miR155-ASO has been reported to alleviate liver injury, efficient drug delivery was a major challenge. RBCEVs loaded with miR155-ASO were specifically delivered to the liver and demonstrated potent therapeutic effects [80]. Despite these noticeable advantages, RBCEVs show some challenges, such as side effects or scalability, to be overcome for further clinical applications [81].

### 2.3. Engineered sEV Therapeutics

sEVs hold tremendous advantages in drug modalities, and many studies have leveraged engineered sEVs to deliver potent macromolecules, including proteins and genes, as shown in Table 4. Due to phospholipid bilayer membranes, EVs provide the optimal microenvironment to therapeutic proteins and allow proteins to perform their original functions as if on cell membranes [82]. Furthermore, therapeutic proteins have been shown to cluster and be enriched in the lipid rafts of sEV membranes, leading to high avidity of the targeted ligand [83]. Therefore, the expression levels of therapeutic proteins are positively correlated with their therapeutic efficacies. Many researchers and biomedical companies have created diverse engineering strategies to effectively display proteins of interest on the surfaces of sEVs to maximize the therapeutic effects [84,85]. For example, Evox Therapeutics found that the genetic engineering of sEV-producing cells to use an N-terminal fragment of syntenin (a cytoplasmic adaptor of syndecan) enhanced the expression efficiency and therapeutic activity of proteins [86]. They created sEVs expressing decoy receptors of inflammatory cytokines to ameliorate inflammatory diseases, such as neuroinflammation, intestinal inflammation, and systemic inflammation. Codiak Biosciences suggested one scaffold protein, PTGFRN (prostaglandin F2 receptor negative regulator), preferentially sorted into sEVs to facilitate the high-level surface expression of proteins of interest [87]. They presented the preclinical and clinical data of sEVs expressing IL-12 via PTGFRN for cancer treatment, verifying the development of a promising technology platform. Recently, Kai Hu et al. observed that the presentation of antigens on sEVs utilizing the transmembrane domain of viral glycoproteins efficiently increased antigen-specific humoral and cellular immunity [88]. Although there are still limitations in expressing a controlled number of proteins on heterogenous EV populations, these efforts represent a substantial advance toward realizing the full therapeutic potential of sEVs.

The intracellular delivery of therapeutic cargo, including genes and proteins, is often unable to surpass the cell membrane. The usage of lipid nanoparticles (LNPs) received attention after the COVID-19 pandemic due to LNPs containing mRNA passing the lipid bilayer of the cell membrane [89]. However, LNPs are artificial nanoparticles that can provoke unexpected immune responses in the body and engender significant long-term toxicity. Moreover, LNPs cannot evade the endolysosomal pathway, which can quickly degrade therapeutic cargo [90]. Surprisingly, a previous study demonstrated that 30% of sEVs evaded the endolysosomal pathway to transfer their cargo into the cell cytosol through a mechanism involving fusion between exosomal and endosomal membranes [91,92]. This unique property is one profound advantage of using sEVs for the intracellular transfer of macromolecules, making undruggable targets druggable. However, the major obstacle to sEV-based intracellular delivery is that efficient methods of loading cargo into sEVs are still under development. Some researchers and biopharmaceutical companies have suggested exogenous loading strategies, including sonication, extrusion, surfactant treatment, dialysis, freeze–thaw treatment, or electroporation, but these methods show very low efficiency. Consequently, endogenous loading strategies using the genetic engineering of EV-producing cells have emerged as alternatives to these methods [93]. For example, Tian et al. reported that incubating CD4^+^ T cell-derived sEVs with anti-VEGF antibodies can suppress angiogenesis and inflammation on choroidal neovascularization [94]. The application of adipose- and stem cell-derived EVs primed by IFNγ helped to repair tendon injury [95]. EVs with transfection of IL-10 overexpressing vector plasmids can modulate T cell immunity in autoimmune uveitis [96]. The chemical engineering of EVs to express a collagen-binding peptide could reduce inflammation and induce muscle regeneration in ischemic disease [97]. The incubation of EVs with anti-angiogenic peptide KV11-anchoring peptide CP05 with EVs could suppress neovascularization in the retina [98]. The loading of therapeutic reagents in sEVs by transfection, incubation, and sonication is widely being tried to utilize EVs as carriers of therapeutic cargo. The loading of paclitaxel into EVs using sonication or electroporation could suppress the neoplastic effect in pulmonary metastasis [99], as well as in Lewis lung carcinoma in an in vivo model [100] and in breast cancer [66], and it prevented alveolar bone loss in periodontitis [101]. The transfection of miRNA containing plasmid after EV isolation could ameliorate diabetic wounds [102], acute lung injury [103], or prevent tumor growth in the brain tumor model [104] and the A549 non-small-cell lung cancer model [105]. However, further research is required to develop efficient methods for the preferential sorting of cargo into sEVs and the loading of the detached cargo from the membranes of sEVs.

### 2.4. sEV Therapeutics in Clinical Trials

Since sEVs can recapitulate the comprehensive therapeutic potential of the donor cell, clinical trials utilizing MSC-derived sEVs are being extensively researched to evaluate the safety of treatment and efficacy on various diseases. The therapeutic dosage widely ranges from 1.2 × 10^12^ to 1.22 × 10^6^ sEV particles per injection, and sEV source cells are diverse, such as adipose-MSCs, bone-marrow-MSCs, and synovial fluid-MSCs.

Recently, sEV treatments for COVID-19 and its complications are also being tested. Since most complications involve respiratory diseases, such as pneumonia and acute respiratory distress syndrome, not only intravenous injection but also the inhalation of sEVs is actively being tested (NCT04969172, NCT04389385, NCT04276987, NCT04747574, NCT04602104). Hitherto, EVs across diverse cellular origins are undergoing clinical trials for a wide range of diseases—from non-life-threatening hair loss to complex neoplasm diseases such as cancer, as shown in Table 5.

**Table 4 pharmaceutics-15-00325-t004:** Therapeutics using engineered sEVs.

Strategy	Source cell	sEV Purification and Engineering	Loaded Cargo Amount	Disease	Therapeutic Schedule	Result	Reference
Surface engineering of sEVs	Mouse spleen-derived CD4^+^ T cells	Ultracentrifugation and incubation with anti-VEGF	10 anti VEGF molecules per sEV	Choroidal neovascularization	10 μg sEVs, intravitreal injection per eyes	Suppressed angiogenesis and inflammation	[94]
	Adipose-derived stem cells	Ultracentrifugation and primed with 100 ng/mL IFNγ overnight	N/A	Tendon injury	EV-laden collagen sheet containing 5–6 × 10^9^ EVs applied around the injury site	Ameliorated the repair site’s inflammatory response, regenerated tendon matrix	[95]
	Human umbilical cord MSCs	Ultracentrifugation and transfection of IL-10-overexpressing vector plasmids	N/A	Autoimmune uveitis	50 μg sEVs, intravenous injection on day 11 post-immunization	Modulated T cell immunity	[96]
	Human placental MSCs	Ultracentrifugation and expression of collagen-binding peptide using click chemistry	N/A	Ischemic disease	1 × 10^10^ sEVs, intramuscular injection into four different sites around the hind limb ischemic region	Reduced inflammation and increased muscle regeneration	[97]
	Human umbilical vascular endothelial cells	Ultracentrifugation and incubation with anti-angiogenic peptide KV11-anchoring peptide CP05	83.1% EVs anchoring KV11 peptides	Retina neovascularization	50 μg sEVs, retro-orbital injection on day 12	Suppressed neovascularization	[98]
Loading cargo into sEVs	Mouse bone marrow-derived macrophages	ExoQuick-TC™ Kit and incubation and sonication with Paclitaxel	10 mg/mL Paclitaxel mixed with 10^11^ sEVs	Pulmonary metastases	4 × 10^11^ particles of sEVs, systemic injection, three times on day 1, 4, and 7	Suppressed neoplastic effect and increased survival period	[99]
	Rat marrow stromal cells	ExoQuick-TC™ Kit and transfection of miR-67 or 146b plasmid	N/A	Brain tumor	50 μg sEV, intratumoral injection on day 5	Reduced glioma growth	[104]
	Human adipose-derived stem cells	Ultracentrifugation and loading of miR-21-5p into sEVs by electroporation	N/A	Diabetic wound	5 μg/200 μL solution applied to the wound bed every 3 days for 15 days	Ameliorated diabetic wounds	[102]
	HEK293	exoEasy Maxi Kit and incubation for loading of curcumin into sEVs	1 μg curcumin in 2.04–2.46 × 10^9^ sEVs	Acute lung injury	15 μg sEVs, intratracheal instillation into the lungs	Reduced proinflammatory cytokines	[103]
	A549	Centrifugation and transfection of TAT and TAR-miR-449a plasmids	N/A	Non-small-cell lung cancer	2 mg/kg sEVs, intravenous injection	Suppressed tumor growth	[105]
	RAW 264.7	ExoQuick-TC™ Kit and incubation, electroporation, and sonication with Paclitaxel	N/A	Lung metastasis	1 × 10^7^ particles, intravenous injection every other day, 7 times	Anticancer effect in Lewis lung carcinoma	[100]
	RAW 264.7	Ultracentrifugation and incubation, electroporation, and sonication with catalase	1376 ± 64.1 U of catalase activity in 4 × 10^11^ sEV/mL	Parkinson’s disease	1.2 × 10^9^ sEVs, injection into each nostril, 10 times every other day	Reduced microgliosis and protected neurons	[106]
	RAW 264.7 M1 macrophage	Ultracentrifugation and sonication with Paclitaxel	N/A	Breast cancer	0.1 mg sEVs, intravenous injection every 3 days for 27 days	Anti-tumor effects	[66]
	Mouse bone marrow-derived dendritic cells	Ultracentrifugation and sonication with Paclitaxel	5 μg TGFB1 and 5 μg IL10 in 1 × 10^9^	Periodontitis	200 µL sEVs, intravenous or palatal tissue local injection on day 3	Prevented cytokines from proteolytic attack and alveolar bone loss	[101]

**Table 5 pharmaceutics-15-00325-t005:** sEV therapeutics in clinical trials.

Applied EVs	Diseases	Dosing Schedules	Expected Results	Patients	Phase	Recruitment Status	Identifier
Amniotic MSC-sEV	Hair loss, alopecia	10^12^ sEVs administered through an interval of 14 days over two months	Anticipation of growth factors contained in stem cell-sEV to improve hair loss	20	N/A	Recruiting	NCT05658094
Placenta MSC-sEV	Complex anal fistula	N/A	Anticipation of anal fistula treatment effect using stem cell sEVs	80	Phase 1, Phase 2	Recruiting	NCT05402748
MSC-sEV	Cerebrovascular disorders	N/A	Evaluation of improvement in acute ischemic stroke patients receiving MSC-sEVs	5	Phase 1, Phase 2	Unknown	NCT03384433
Embryonic kidney T-REx™-293 cell-sEV	COVID-19	10^10^ sEVs/4 mL normal saline administered through inhalation, once a day for 5 days	Evaluation of the safety and efficacy of sEVs overexpressing CD24	155	Phase 2	Active, not recruiting	NCT04969172
Adipose tissue stem cell-sEV	Periodontitis	N/A	Evaluation of regeneration effect	10	Early Phase 1	Unknown	NCT04270006
Plasma-sEV	Ulcer	N/A	Anticipation of skin wound-healing efficacy of plasma sEVs	5	Early Phase 1	Unknown	NCT02565264
sEV	Neuralgia	N/A	Evaluation of safety and efficacy of sEVs in patients with craniofacial neuralgia	100	N/A	Suspended	NCT04202783
MSC-sEV	Myocardial infarction, myocardial ischemia, myocardial stunning	100 μg/mL sEVs administered through intracoronary and intra-myocardial injection	Anticipation of improvement in patient outcomes from the detrimental effects of ischemia and reperfusion injury	20	Phase 1, Phase 2	Recruiting	NCT05669144
sEV	Refractory depression, anxiety disorders, neurodegenerative diseases	2.1 × 10^7^ allogenic sEVs/15 mL administered through intravenous injection	Evaluation of efficacy of sEVs in patients with neurodegenerative dementia	300	N/A	Suspended	NCT04202770
MSC-sEV	SARS-CoV-2 infection	sEV administered through intravenous injection twice, in day 1 and day 7 of 2 weeks	Evaluation of efficacy of MSC-sEVs in reducing hyper-inflammation in patients with moderate COVID-19	60	Phase 2, Phase 3	Recruiting	NCT05216562
MSC-sEV	Multiple organ failure	180 mg sEV administered through intravenous injection once a day for 14 days	Evaluation of efficacy of MSC-sEVs for multiple organ dysfunction syndrome	60	N/A	Not yet recruiting	NCT04356300
Mesenchymal progenitor cell-sEV	Drug-resistant	(8 or 16) × 10^3^ sEVs/3 mL administered through inhalation 7 times, day 1 to 7	Evaluation of efficacy of sEV treatment for pulmonary infection caused by carbapenem-resistant gram-negative bacilli	60	Phase 1, Phase 2	Recruiting	NCT04544215
Wharton jelly MSC-sEV	Retinitis pigmentosa	sEVs administered through single subtenon injection	Evaluation of efficacy of umbilical cord MSC-sEVs in the treatment of retinitis pigmentosa	135	Phase 2, Phase 3	Recruiting	NCT05413148
COVID-19-specific T cell-sEV	Corona virus infection pneumonia	2.0 × 10^3^ sEVs/3 mL administered once a day for 5 days	Evaluation of efficacy after targeted delivery of T cell sEVs by metered-dose inhaler	60	Phase 1	Unknown	NCT04389385
MSC-sEV	Psoriasis	100 µg MSC sEV/g of ointment was dripped once a day for 20 days	Anticipation of psoriasis treatment efficacy using MSC-sEV ointment	10	Phase 1	Completed	NCT05523011
Adipose MSC-sEV	Corona virus	2.0 × 10^8^ sEVs/3 mL administered through aerosol inhalation 5 times for 5 days	Anticipation of safety and efficacy of stem cell sEVs in treatment of severely ill patients hospitalized with novel coronavirus pneumonia	24	Phase 1	Completed	NCT04276987
MSC-sEV	COVID-19, novel coronavirus pneumonia, acute respiratory distress syndrome	(2, 4, and 8 × 10^9^ or 8, 4, and 8 × 10^9^) sEVs/mL administered through injection every other day for 5 days or 8 × 10^9^ sEV administered through injection every other day for 5 days	Anticipation of treatment effects of stem cell sEVs in treatments of patients with acute respiratory distress syndrome and novel coronavirus pneumonia	55	Phase 1, Phase 2	Not yet recruiting	NCT04798716
Neonatal stem cell-sEV	Neuralgia	N/A	Anticipation of treatment in neuralgia patients using neonatal sEVs	100	N/A	Suspended	NCT04202783
MSC-sEV	Osteoarthritis	(3–5) × 10^11^ sEVs administered through single dose injection	Anticipation of knee arthritis treatment effect using stem cell sEVs	10	Phase 1	Not yet recruiting	NCT05060107
T-REx™-293 cell-sEV	SARS-CoV-2	(1, 5, 10, 100) × 10^8^ sEVs/2 mL saline) administered through QD inhalation device	Anticipation of therapeutic effect in moderate/severe COVID-19 patients with sEV-CD24	35	Phase 1	Recruiting	NCT04747574
BM MSC-sEV	Familial hypercholesterolemia	(0.044, 0.088, 0.145, 0.220, 0.295, or 0.394 mg/kg) sEV administered through single dose injection	Anticipation of treatment of hypercholesterolemia patients with sEV-based LDLR mRNA nano platform	30	Phase 1	Not yet recruiting	NCT05043181
Synovial fluid MSC-sEV	Degenerative meniscal injury	1 × 10^6^ sEVs/kg administered through intra-articular injection	Anticipation of efficacy of synovial MSC-sEVs in patients with degenerative meniscal cartilage damage	30	Phase 2	Recruiting	NCT05261360
Cell free umbilical cord-blood-MSC-sEV	Diabetes mellitus type 1	(1.22–1.51) × 10^6^ sEVs/kg administered through intravenous injection	Anticipation of cord blood-sEVs to treat type 1 diabetes patients	20	Phase 2, Phase 3	Unknown	NCT02138331
Umbilical MSC-sEV	Dry eye	10 ug/drop sEV administered through eye drops 4 times a day for 14 days	Anticipation of MSC-sEVs to alleviate dry eye symptoms in patients with chronic graft versus host diseases	27	Phase 1, Phase 2	Recruiting	NCT04213248
Adipose-sEV	Wounds and injuries	N/A	Anticipation of sEVs to promote wound healing	5	N/A	Not yet recruiting	NCT05475418
MSC-sEV	COVID-19 SARS-CoV-2 pneumonia	(0.5–2 × 10^10^) sEVs of the first or second type/3 mL administered through inhalation twice a day for 10 days	Anticipation of safety and efficacy of sEV inhalation method for COVID-19-associated pneumonia	90	Phase 2	Unknown	NCT04602442
sEV	COVID-19	(1 or 10) × 10^9^ sEVs administered once a day for 5 days	Anticipation of safety and efficacy assessment of CD24 overexpressing sEVs for patients with severe COVID-19	90	Phase 2	Recruiting	NCT04902183
MSC-sEV	Acute respiratory distress syndrome	(2, 8, 16) × 10^8^ sEVs administered through inhalation 7 times 1/4 MTD/day, or MTD/day at day 1 to 7	Anticipation of treatment effects of MSC-sEVs in acute respiratory distress syndrome	169	Phase 1, Phase 2	Unknown	NCT04602104
MSC-sEV	Macular hole	50 or 20 μg/10 μL sEV was dripped into vitreous cavity around macular holes	Evaluation of safety and efficacy evaluation of MSC-sEVs for promoting healing of large and refractory macular holes	44	Early Phase 1	Active, not recruiting	NCT03437759
Adipose MSC-sEV	Alzheimer’s disease	(5, 10, or 20 μg/1 mL) sEV administered through nasal drip twice a week for 12 weeks	Evaluation of safety and efficacy evaluation of allogeneic adipose MSC-sEVs in patients with Alzheimer’s disease	9	Phase 1, Phase 2	Unknown	NCT04388982
MSC-sEV	Dystrophic epidermolysis bullosa	N/A	Evaluation of the safety and efficacy of sEVs in the treatment of lesions in patients with epidermolysis bullosa	10	Phase 1, Phase 2	Not yet recruiting	NCT04173650
ASO-STAT6-sEV	Advanced hepatocellular carcinoma, gastric cancer metastatic to liver colorectal cancer metastatic to liver	N/A	Evaluation of treatment efficacy for advanced hepatocellular carcinoma and primary gastric cancer with ASO-STAT6 sEVs (CDK-004)	30	Phase 1	Recruiting	NCT05375604
sEV	Cutaneous T-cell lymphoma	N/A	Evaluation of safety, tolerability, pharmacokinetics, and pharmacodynamic effects of CDK-003	2	Phase 1	Terminated	NCT05156229
BM MSC-sEV	COVID-19 acute respiratory distress syndrome	1.2 × 10^12^ sEVs/85 mL, 8 × 10^11^ sEVs/90 mL administered through intravenous injection	Evaluation of safety and efficacy of acute respiratory distress syndrome treatment using DB-0018 SEV	120	Phase 2	Completed	NCT04493242
DC-sEV	Non-small-cell lung cancer	1.3 × 10^13^ MHC class II molecules administered through injection, four doses at weekly intervals	Evaluation of the safety, feasibility, and efficacy of administering tumor antigen-loaded autologous dexosomes to patients with advanced non-small-cell lung cancer (NSCLC)	13	Phase 1	Completed	
Turmeric-sEV	Colon cancer	3.6 g curcumin-conjugated sEV tablets taken daily for 7 days	Evaluation of the ability of sEVs to deliver curcumin more effectively to normal colon tissues and colon tumors	7	Phase 1	Active, not recruiting	NCT01294072

## 3. Issues to Overcome for Realization of sEV Therapeutics

### 3.1. Biodistribution

An sEV biodistribution (BD) evaluation should be conducted to create a new, effective class of medicines and to begin the first in-human studies. Some questions still need to be answered regarding the BD of sEVs. For example, how do the different administration routes affect sEVs’ BD? What is the best labeling method for sEVs? To date, the most popular administration route for sEVs in preclinical studies is intravenous injection, occupying more than half of the total [107]. Much evidence has shown that the primary accumulation organs of intravenously injected sEVs are the liver and spleen, the reticuloendothelial systems [107]. However, studies comparing the efficacy of sEVs’ BD using different injection routes are scarce [108]; therefore, further studies of sEVs’ BD for the delivery of therapeutic cargo using sEVs to specific organs, including the lungs and brain, are required.

Despite many studies attempting to accurately assess the BD of sEVs using diverse labeling methods, the gold standard for labeling EVs has yet to be determined. The most widely utilized labeling approach is lipophilic fluorescent dye, including PKH, and diakylcarbocyanine dyes (DiD, Dil, Dio, and DiR), which can be readily integrated into the membranes of sEVs. However, these lipophilic dyes may aggregate sEVs and cause background/pseudo signals. Moreover, these dyes eventually affect the composition of the surfaces of sEVs, leading to effects on the biological activity of sEVs [109]. Several other methods have been attempted, such as encapsulation, metabolic labeling (e.g., click chemistry), and surface modification by genetic engineering (luciferase). However, these approaches showed limitations in measuring the accurate pharmacokinetics (PK) of sEVs [109]. Some papers have shown a covalent binding method using Cy dyes (Cy5.5 and Cy 7) or radioisotopes (64Cu, 68Ga, 125I, 99mTc, and 89Zr) to sEV surfaces [110]. Although this covalent binding may also affect the interaction of sEVs and targeted cells, this method has a low risk of eliciting pseudo-signals caused by the release of free-from dyes from dye-conjugated sEVs [110]. Moreover, radioisotope-based imaging with positron emission tomography (PET) and single-photon emission computed tomography (SPECT) were shown to be highly sensitive over in vivo tracking compared with fluorescence or luminescence imaging [111]. Taken together, we can begin to understand the BD and PK of sEVs with recent advancements in labeling methods, but further work is required to reveal accurate data.

### 3.2. Large-Scale Manufacturing

Compared to conventional therapeutics such as protein drugs, antibodies, and cell or gene therapeutics, there is no state of the art for the large-scale production of EV products, and there is also no concrete regulation or guidance from a regulatory board such as the FDA or EMA. Nonetheless, current EV therapeutics are actively being developed; thus, the establishment of manufacturing protocols and regulatory guidelines is needed. Since EVs are retrieved from naïve or engineered cells, the overall production process is perceived as similar to that of cell or gene therapy products. The master cell banking process is required to collect sEVs to maintain the cell homology, such as surface molecule expression, intracellular content, and engineered traits. In terms of engineered EVs, ex vivo manipulation is the leading strategy. For instance, the cells can be transduced with a retrovirus, adenovirus, or lentivirus to express the desired EVs stably [112]. The transduced stable cells are stored and managed in cryopreserved form, often referred to as a master cell bank (MCB), which can be utilized to assess product quality. EV manufacturing processes can be divided into two main types: upstream process development (USP) and downstream process development (DSP).

The MCBs or banked source cells are used for USP. The banked cells are thawed and expanded through the culture process. Cells are seeded on an appropriate culture dish, depending on the cell type. Alternative culture systems, such as 3D fiber cell systems, cell stacks, or seed trains, are used to increase the production of sEVs per cell, since these platforms improve cell viability and enable high-density cell growth. After adequate expansion, cells are transferred to a 3D bioreactor, an automated system optimized for cell growth. Once the cells are fully expanded, the culture medium is exchanged for serum-free media to inhibit soluble protein contamination and secure the purity of the final product. During downstream process development, serum-free medium is collected, and purified EVs are isolated. There are no set standard procedures, but most DSP resembles that of cell or gene therapeutic development. DSP focuses on collecting high-purity EVs with desired yields, appropriate for commercialization. Since the yield of the product and purity are trade-offs, the key in DSP development is to optimize both variables for a high-quality product with a practical outcome. The initial step is to remove the potential contaminants and collect small-sized EVs by depth filtration. Serial filtration, or depth filtration, sorts desired EVs from non-desired EVs through size cut-offs. This step is similar to the serial centrifugation process during the lab-scale production of EVs. Once the filtered EVs are retrieved, the product undergoes tangential flow filtration (TFF) to minimize the damage of EVs, maximize the purity, and concentrate the media into higher concentrations. The concentrated product undergoes a chromatography step to enhance the purity. Different types of chromatography columns are used, such as size exclusion chromatography or ion exchange chromatography, depending on the physical and chemical features of the EVs. Since most chromatography steps result in the dilution of the samples, the retrieved EVs often undergo TFF once more for concentration. The final drug products are packaged through fill-and-finish procedures. Though the gold standard of this step is also not yet established, many CDMOs and biopharmaceuticals lyophilize acquired EVs for higher stability and to facilitate storage and transport.

### 3.3. Quality Control

Quality control (QC) tests are crucial for the clinical translation of sEV therapeutics. Protein- and small-chemical-based medicines should be verified as a homogenous population through several robust QC tests. However, the heterogeneous populations of sEVs cannot be converted into a homogenous population. Instead, the batch-to-batch consistency demonstrated by appropriate QC tests is the major priority for the GMP-grade manufacturing of sEV therapeutics. To prove the batch-to-batch consistency, we must establish a list of QC tests on final products (sEVs) with a sufficient scientific rationale to persuade the FDA or other regulatory authorities to approve human clinical trials. The International Society for Extracellular Vesicles (ISEV) suggested minimal requirements in the MISEV2018 guidelines for the quality control of sEVs [1]. According to their guidelines, QC items of sEVs include the quantification (particle number, protein, and lipid), size (<200 nm), identification (the positive and negative markers), and purity (ratios of proteins or lipids: particles or proteins: lipids).

With the advancement of single-particle analysis technologies, quantifying the amount of therapeutic cargo loaded into a single particle is becoming feasible. Developed by NanoView Biosciences, the Exoview R200 automatically analyzes the EVs’ number and size data through probed tetraspanin markers, including CD63, CD81, and CD9, by taking micro-biochip-based fluorescent images [113]. Furthermore, by customizing the biochip according to the experimenter’s needs, the amount of therapeutic cargo loaded in EVs can be measured for each particle. Similarly, NanoAnalyzer, developed by NanoFCM, is a device that distinguishes between sEVs with a diameter of 40 nm and protein aggregates, enabling the evaluation of the characteristics of single sEV using antibodies, such as flow cytometry, which analyzes cells [114]. NanoAnalyzer is cost-effective, since additional consumables other than the device are not required. Moreover, it is familiar to users due to the interface’s similarity to existing flow cytometry methods. These devices for single EV analysis can assess not only the shape and size of a single EV, but also EV markers, enabling the more accurate measurement of the purity of sEVs. These advanced developments in QC tests are rapidly accelerating the clinical realization of sEV therapeutics. Therefore, an essential task is to develop a rigorous assay suitably customized for each sEV therapeutic.

## 4. Conclusions and Perspectives

sEVs are naturally produced in our bodies and play vital roles in biological functions, with numerous advantages as a new class of medicines. Developing platform technologies and establishing therapeutic strategies that maximize the advantages of sEVs are considered the most significant paradigm shifts in creating new treatments. Although sEV therapeutics have not yet been approved and used in patients, numerous clinical trials based on sEVs have recently been attempted, and the numbers are constantly increasing. The large-scale manufacturing and QC of sEVs, which were previously inconceivable, have also made much progress in convincing regulatory authorities. There are still issues to be solved, but we expect that continuous technological development and research will establish innovative sEV-based treatments as promising therapeutic options to solve existing high unmet medical needs.

## Figures and Tables

**Figure 1 pharmaceutics-15-00325-f001:**
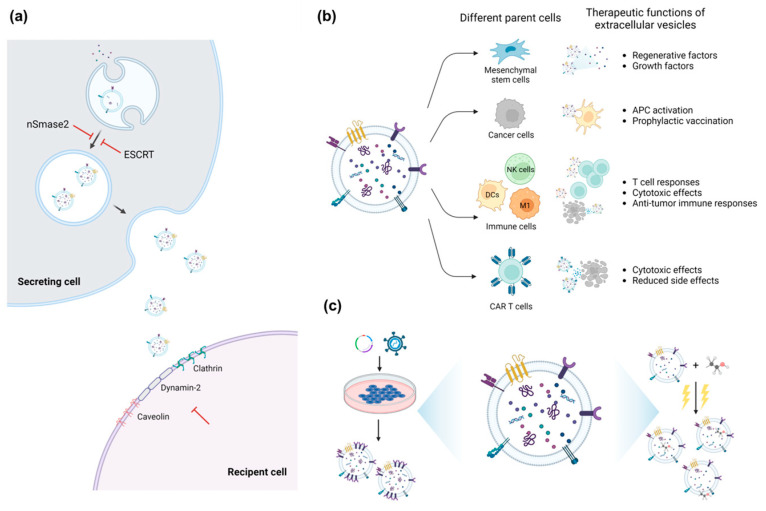
Types and methods of EV-mediated therapeutics. (**a**) Inhibition of sEV release of secreting cells or sEV uptake by recipient cells can be utilized to prevent the progression of diverse diseases. (**b**) sEVs show diverse therapeutic functions depending on the functionality of source cells. (**c**) sEVs can be engineered before and/or after isolation from the cell media. The most well-known methods are the transfection of engineered vectors to embody desired traits or electroporation to load therapeutic proteins into isolated sEVs.

**Table 2 pharmaceutics-15-00325-t002:** Therapeutics using stem cell-derived naïve sEVs.

Source Cell	sEV Purification	Disease	Therapeutic Schedule	Result	Reference
Mouse bone marrow MSCs	Ultracentrifugation	Acute myocardial infarction	50 μg MSC-sEVs in 25 μL PBS, intramyocardial injection	Changed M1 macrophages to M2 by delivery of miR-182	[33]
Human liver stem cells	Ultracentrifugation, purification by iodixanol	Chronic kidney disease	1 × 10^10^ particles/mL MSC-sEVs, intravenous injection weekly for 4 weeks	Anti-fibrosis and improvement in kidney function	[36]
Human bone marrow MSCs	Filtration, Total Exosome Isolation Reagent	GVHD	2 × 10⁶ particles/kg MSC-sEVs in 200 μL saline, intravenous injection	Immunomodulatory effects on T cells by delivering miRNA	[38]
Rat urine stem cells	Ultracentrifugation, purification by 30% sucrose/D_2_O cushion	Diabetic nephropathy	100 μg MSC-sEVs in 200 μL of PBS, intravenous injection weekly for 4 weeks	Ameliorated kidney impairment	[37]
Canine adipose tissue MSCs	Ultracentrifugation	Inflammatory bowel disease	100 µg MSC-sEVs from either naïve or primed cASCs, in 200 µL PBS, intraperitoneally injected at days 1, 3, and 5	Increased the immune modulatory effect	[39]
Human bone marrow MSCs	Ultracentrifugation	Tumor	First, 100 μg MSC-sEVs; followed by 50 μg MSC-sEVs, in a volume of 20 mL of PBS, intravenous injection weekly for 4 weeks	Inhibited tumor growth	[40]
Human bone marrow MSCs	100 kDa ultra-filtration, Exo Quick-TC^TM^ Kit	Retinal ischemia	4 μL of 1 × 10^9^ particles/mL MSC-sEVs, intravitreous humor injection	Neuroprotection and regeneration	[41]
Human umbilical cord MSCs	Ultracentrifugation	Eye subretinal fibrosis	2 μL MSC-sEVs, intravitreous tumor injection	Ameliorated subretinal fibrosis by delivering miR-27b	[42]
Human umbilical cord blood MSCs	Ultracentrifugation	Choroidal neovascularization	1 μg, 2 μg, 3 μg of 50 μg/Ml MSC-sEVs, intravitreal humor injection	Ameliorated RPE cells and retina via downregulation of VEGF-A	[43]
Mouse neural stem cells	Ultracentrifugation	Huntington’s disease	10 μL of 5 mg/mL MSC-sEVs, injection into area between the first and second lumbar vertebrae, twice after a 7-day interval each	Reduced mutant HTT aggregation in the brain	[44]
Wharton’s jelly	Ultracentrifugation, Exo-Prep kit	Alzheimer’s disease	50 µg MSC-sEVs, intravenous injection, weekly for 4 weeks	Downregulated HDAC4, improved AD pathology	[45]
Human bone marrow MSCs	Centrifugation, purification by PEG solution	Alzheimer’s disease	2 × 10^9^ MSC-sEVs in 5 µL, intranasal injection every 4 days for 4 months	Improved in cognitive tests	[34]
Human umbilical cord blood MSCs	Exo Quick-TC^TM^ kit	Alzheimer’s disease	30 μg MSC-sEVs, intravenous injection, every 2 weeks, four times	Reduced neuroinflammation and Aβ deposition	[46]
Murine bone marrow MSCs	Ultracentrifugation	Alzheimer’s disease	5 × 10^11^ MSC-sEVs, intravenous injection, monthly for 4 months	Lessened plaque deposition, restored inflammatory cytokine levels	[47]
Human bone marrow MSCs	Ultracentrifugation	Alzheimer’s disease	100 μg MSC-sEVs, intracerebroventricular injection, once every 2 days for 2 weeks	Reduced iNOS expression, relieved synaptic impairment, and long-term potentiation	[48]
Mouse bone marrow-derived MSCs	ExoQuick	Alzheimer’s disease	150 μg MSC-sEVs, intravenous injection, biweekly for 4 months	Recovered learning and memory capabilities and synaptic dysfunction	[49]
Mouse bone marrow MSCs	Ultracentrifugation	Alzheimer’s disease	22.4 μg MSC-sEVs, single intracerebral injection	Ameliorated Aβ burden and dystrophic neurites	[50]
Human umbilical cord MSCs	Ultracentrifugation	Alzheimer’s disease	2 mg/mL intracerebroventricular injection	Reduced Aβ generation and oxidative stress, prevented microglia activity	[51]
Human umbilical cord MSCs	Centrifugation	Alzheimer’s disease	5 × 10^5^ MSC-sEVs, intravenous injection, at weeks 2 and 3	Improved neurogenesis and neuroinflammation properties	[51]

**Table 3 pharmaceutics-15-00325-t003:** Therapeutics using immune cell-derived naïve sEVs.

Source Cell	sEV Purification	Disease	Therapeutic Schedule	Result	Reference
Murine bone marrow-derived DCs	Ultracentrifugation	Mastocytoma, mammary adenocarcinoma	3–5 μg sEVs, single intradermal injection	Suppressed tumor growth, primed tumor-specific cytotoxic T cells	[70]
NK-92MI human NK cells	Ultracentrifugation	Melanoma	20 μg sEVs, intertumoral injection for two days	Suppressed tumor growth	[64]
Peripheral blood mononuclear cell-derived T cells	Ultracentrifugation	Triple-negative breast cancer	240 μg sEVs, intraperitoneal injection, every 3 days for 27 days	Decreased tumor cell-induced T cell dysfunction	[71]
NK cells	Ultracentrifugation	SK-N-SH neuroblastoma	Treated with sEVs, intraperitoneal injection, repeated three times with a 7-day interval until death	Increased survival time	[65]
NK cells	Ultracentrifugation	Anti-tumor effect	100 µg sEV injection	Suppression of tumor growth	[72]
CAR-T cells	Ultracentrifugation	MDA-MB-231, HCC827, SK-BR-3	25–125 µg sEV injection, every week for 40 days	Suppression of dose-dependent tumor growth	[69]
CAR-T cells	Ultracentrifugation	Breast cancer	100–500 μg sEVs, intravenous injection on days 3, 6, 9, 12, and 15	Inhibited tumor growth, low toxicity	[73]

## Data Availability

The data presented in this study are available in this article.

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
