# Peer review of "Small Extracellular Vesicles as a New Class of Medicines"

_pharmaceutics, 2023, doi:10.3390/pharmaceutics15020325_

Round 1
Reviewer 1 Report
Dear Authors and Editors! I have some comments to the manuscript.
1) My opinion is that the use of "harnessing" in the title should be avoided since this is not very common word for the non-native speakers. This may be crucial for the amount of citations the review will generate.
2) I suggest the authors should add the chapter devoted to the particular properties of sEV which make them possible to be used as therapeutics, before the chapter 2 "Types of sEV Therapeutics". Also differences between sEV and specific subfractions, i.e. exosomes should be described.
3) The data listed in the Table 1 should be discussed in the text in more details.
4) The title of the Chapter 2 is "Types of sEV Therapeutics", but the subchapters are more about sources of sEV and strategies of use of sEV. So apperently the structure of the manuscript should be:
2. Strategies of use
2.1. Inhibition of the release and uptake of exosomes
3. Sources of sEV
2.2. Stem cell-derived naïve sEV
2.3. Immune cell-derived naïve sEV
2.4. Engineered sEV
Since there are a plenty of other sources of sEV (and exosomes), which have potential clinical use, these sources also should be reviewed in the paper.
5) The review is devoted to the medicines, so the Table with the list of sEV-related drugs in clinical trials (according to the clinicaltrials.gov) is appreciated, containing the therepautic targets and stage of the trials.
Author Response
Point-by-point responses to the reviewers’ comments on
“Small extracelluar vesicles as a new class of medicines”
We sincerely appreciate the time and effort of editor and each reviewer. Reviewers’ comments and insightful feedbacks have indeed bolstered our paper. Please note that our paper underwent extensive English revisions for better legibility through the editing service listed on https://www.mdpi.com/authors/english. We have addressed each comment of the referees as a separated file. Once again, thank you for rigor feedbacks and investing your time in our paper.

Reviewer 2 Report
In this review manuscript, an elaborate summary of recent progress in using extracellular vesicles (EVs) as therapeutics was presented. This is an interesting topic, and using EVs as/for therapeutic delivery is an important and meaningful research field. The authors summarized different types of sEV therapeutics, future improvements, and current strategies of large-scale production, which should be helpful for researchers to understand the importance and the latest progression of EVs for drug delivery. The authors have also made effective use of figures and tables. Thus, I recommend the acceptance of this manuscript after considering the following points.
1. In the introduction, a disclaimer of sorts about the proper usage of terms is given as an opening paragraph. I suggest it be moved to the second paragraph just for clarity of the manuscript.
2. A short introduction of other types of EVs such as microvesicles (MVs) and apoptotic bodies should be supplemented and more explanation on choosing sEVs as the representatives of EVs should be provided.
3. line 74- exosomes or sEVs? Please use one term throughout the manuscript.
4. In 2.1, giving a few examples of the adverse effects of sEVs would help readers understand the importance of inhibiting release or uptake before jumping right into therapeutic interventions.
5. Lines 236-237 – this claim is missing some important pioneering references (10.1021/acsnano.9b10033, 10.1128/JVI.01578-18)
6. Section 3 is very well written. Important considerations for advancing the field.
7. Lines 271, 276 – [REF]? please add the references.
8. Lines 283-297 – a very similar concept was recently put forth in another review. Please refer to it (10.1016/j.mtnano.2021.100148).
9. This is a bit of work, however, the cargo loaded and loading efficiency are important parameters to add in Table 4. The authors are suggested to provide available data on loading efficiency and the cargo loaded for all references.
10. In Tables 1 and 4, it is difficult to know which accounts belong to which strategy as there are no separation lines. Please add them.
Author Response

(The authors gave the same response as above.)

Round 2
Reviewer 1 Report
The manuscript has been significantly improved compared to the previous version. I recommend native speaker language polishing or the use of the MDPI language editing service.
Sincerely